# Minerva: Enhancing Quantum Network Performance for High-Fidelity Multimedia Transmission

## ABSTRACT

Quantum networks have the potential to transmit multimedia data with high security and efficiency. However, ensuring high-fidelity transmission links remains a significant challenge. This study proposes a novel framework to enhance quantum network performance via link selection and transport strategy. Specifically, we formalize the quantum fidelity estimation and link selection as a best-arm identification problem and leverage median elimination to estimate fidelity and select the quantum link for each multimedia chunk transmission. To optimize the transmission of multimedia chunks in a quantum network, we can employ the scheduling strategy to maximize the cumulative benefit of chunk transmissions while considering the fidelity of the links and the overall network utilization. Through extensive experiments, our proposal demonstrates significant advantages. Compared to the randomized method, *Minerva* reduces bounce number and execution time by 12% ∼ 28% and 8% ∼ 32%, respectively, while improving average fidelity by 15%. Compared with the uniformly distributed method, our approach decreases bounce number by 24% ∼ 30% and execution time by 8% ∼ 32% and enhances average fidelity by 11% ∼ 21%.

## CCS CONCEPTS

• **Information systems** → **Multimedia streaming**; • **Networks** → **Data path algorithms**.

## KEYWORDS

Qauntum network and links, multimedia transmission, multi-armed bandit, quantum fidelity

## 1 INTRODUCTION

Quantum networks herald a new era in networking, facilitating applications that were previously deemed impossible using classical means, achieving unique cryptographic advantages by exploiting fundamental principles of quantum mechanics, such as quantum cryptography [5], quantum teleportation [9], quantum key distribution (QKD) [12], and quantum internet-of-things (QIoT) [25]. In the realm of multimedia [27], the integration of quantum networking holds significant promise for transformative applications. Leveraging the unique properties of quantum communication, such as superposition and entanglement, can revolutionize various aspects of multimedia processing, storage, and distribution [19].

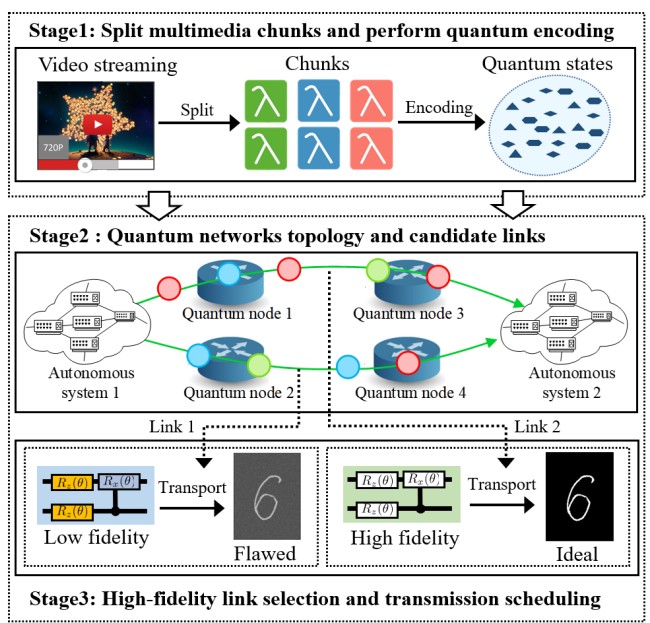

**Figure 1: Illustrative explanation of Minerva framework.**

One compelling application lies in secure multimedia transmission. Quantum encryption techniques, such as QKD, offer unbreakable security protocols by harnessing the principles of quantum mechanics. By utilizing entangled photon pairs to generate cryptographic keys, quantum networks can ensure the confidentiality and integrity of multimedia data during transmission, protecting sensitive information against eavesdropping and cyber threats. Moreover, quantum networks can enhance the efficiency and reliability of multimedia content delivery. Quantum repeaters [17, 24], designed to overcome the inherent limitations of optical fiber transmission, enable the long-distance transfer of quantum states with minimal loss. This capability is particularly advantageous for multimedia streaming services, where high-quality, low-latency transmission is essential. By employing quantum repeaters, multimedia content providers can deliver seamless, high-fidelity streaming experiences to users across global networks [11, 18].

In the context of multimedia transmission, quantum networks offer a novel approach to enhance the security and integrity of data transfer. Figure 2 illustrates a typical pipeline where multimedia data, such as video streams, are first segmented into multiple chunks. These chunks are then subjected to quantum encoding, a process that encodes the information into quantum states for transmission over the quantum network. Given a specific quantum network topology, each quantum state of the data chunk can be transmitted over different links. However, due to the fragility of quantum information, quantum bits (or qubits) can easily decoherence through

interactions with the environment. For example, imperfect entangled pairs and physical operations may lead to transmission failure during the establishment of long-distance entanglement. Typically, in addition to standard metrics like throughput and delay, a key metric is called *fidelity*[1] [20]. Fidelity is a quantum metric with no classical equivalents and is used to quantify the quality of an expected quantum state. To cope with this, it is desirable to transmit over links that maintain high fidelity, while also considering transmission efficiency. In cases where the optimal links with high fidelity are not available or are congested, it may be necessary to utilize suboptimal links with lower fidelity to ensure the continuity of transmission.

In this paper, we propose Minerva[2], to enhance quantum network performance for high-fidelity multimedia transmission. Specifically, we formalize the quantum fidelity estimation and link selection as a best-arm identification problem and leverage median elimination [6] to estimate fidelity and select the quantum link for each multimedia chunk transmission. To optimize the transmission of multimedia chunks in a quantum network, we can employ the scheduling strategy to maximize the cumulative benefit of chunk transmissions while considering the fidelity of the links and the overall network utilization.

In a nutshell, this paper makes three key contributions as follows:

- We have meticulously examined the constraints associated with the application of quantum networks in multimedia transmission, particularly concerning the selection of high-fidelity links and the implementation of dynamic allocation strategies.
- Specifically, we first model the quantum network topology into multiple candidate links, and then we design a tailor-made link selection algorithm for Minerva based on median elimination (which solves the best-arm identification problem). Meanwhile, the profit of each multimedia chunk transmission is modeled to implement a dynamic allocation strategy.
- Through extensive experiments, our algorithm, *Minerva*, demonstrates significant advantages. Compared to the randomized method, *Minerva* reduces bounce number and execution time by 12% ∼ 28% and 8% ∼ 32%, respectively, while improving average fidelity by 15%. When compared with the uniformly distributed method, our approach decreases bounce number by 24% ∼ 30% and execution time by 8% ∼ 32% and enhances average fidelity by 11% ∼ 21%.

## 2 BACKGROUND AND RELATED WORK

### 2.1 Quantum Network

Quantum networks represent an emerging paradigm in the field of quantum information science, leveraging the principles of quantum mechanics to enable secure and efficient communication. Unlike classical communication networks, quantum networks offer the potential for ultra-secure communication protocols, including quantum key distribution (QKD) [12] and quantum teleportation [9], which are immune to eavesdropping and interception due to the inherent uncertainty and non-cloning properties of quantum systems. The architectural framework of quantum networks is designed to facilitate efficient communication among quantum nodes, which form the backbone of the quantum internet. *Quantum nodes*, the fundamental units of a quantum network, utilize quantum communication links to transmit quantum bits (qubits). These links can be optical fibers, free-space optical links, or other quantum channels based on various transmission media. A pivotal feature of quantum networks is the ability to establish connections between distant quantum nodes through quantum entanglement. That is *quantum nodes* process and transmit quantum information via *quantum links*, while *entanglement swapping* extends entanglement across nodes to facilitate long-distance communication.

The key component of quantum network architecture is quantum nodes, which serve as the fundamental building blocks for information processing and transmission. These nodes typically consist of quantum processors capable of manipulating quantum states, quantum memories for storing quantum information, and quantum interfaces for interfacing with external quantum systems. Quantum links [3, 14] form the backbone of quantum communication networks, facilitating the transmission of quantum information between nodes. These links often comprise optical fibers or free-space channels, where quantum states encoded on photons are transmitted over varying distances. However, the fidelity of quantum states transmitted through these links is susceptible to quantum noise (*e.g.,*, decoherence), leading to information loss and degradation. Entanglement swapping [10] is a key technique for long-distance quantum communication within quantum networks. This method enables two quantum nodes to "swap" entangled states through a series of operations and an intermediary node, thereby creating a direct quantum correlation between them. This approach circumvents the limitations of direct qubit transmission, such as losses and noise, and also circumvents the challenges posed by the quantum no-cloning theorem.

The basic quantum mechanical principles include the uncertainty principle, measurement collapse, and the non-cloning theorem. The uncertainty principle prevents eavesdroppers from intercepting quantum information undetected, the measurement collapse enables eavesdropping attempts to be detected, and the principle of non-clonability prevents unauthorized access by prohibiting faithful copying of quantum states. These principles therefore underlie the security mechanisms inherent in quantum communication systems. As quantum networks evolve, researchers have proposed diverse quantum network architectures. For instance, Kozlowski *et al.* [10] proposed a quantum network protocol for end-to-end quantum communication, focusing on efficient entanglement generation. Their approach introduces an Entanglement Generation Switch (EGS), facilitating resource-sharing among multiple quantum nodes. Andrade *et al.* [4] focuses on characterizing channel noise with bit-flip in quantum networks using Network Tomography Protocols (NTP).

---

[1] It has a value between 0 and 1, a fidelity of 1 means that it is in the desired state and a value below 0.5 means that the state is no longer available. Unlike classical networks requiring error-free transmission, quantum applications can function with imperfect states, provided the fidelity surpasses an application-specific threshold (*e.g.,* the threshold fidelity is around 0.8 for basic QKD) [3].

[2] Minerva is the Roman goddess associated with wisdom, justice, victory, and other aspects including arts, trade, and strategy.

## 2.2 Quantum Noise and Average Fidelity

Current quantum computing is in the noisy intermediate-scale quantum (NISQ) era and noise is an intrinsic feature of quantum computing [21]. Quantum noise arises from various sources, including thermal fluctuations, electromagnetic radiation, and material imperfections in the quantum hardware. These interactions introduce fluctuations that couple the quantum system to its surroundings, resulting in the entanglement of the system with the environmental states and the subsequent loss of quantum coherence. Quantum noise poses a significant challenge in quantum communication, leading to the loss of quantum coherence and degradation of fidelity. Arising from interactions with the environment, quantum noise causes quantum states to become entangled with their surroundings, rendering them susceptible to information loss and corruption.

In quantum networks, fidelity serves as a critical metric for assessing the quality of entanglement between nodes in quantum communication, quantifying the similarity between an actual quantum state and its desired target state. It ranges from 0 to 1, with a fidelity of 1 indicating perfect alignment with the target state. However, in practice in the NISQ era, quantum states are susceptible to defects and errors due to noise, resulting in fidelity values lower than 1, and fidelity below 0.5 is generally considered unavailable. Therefore, in order to ensure the realization of efficient transmission of quantum information, the estimated quantum fidelity is usually required to be above a certain threshold value, for instance, the threshold fidelity is around 0.8 for basic QKD [3].

Average fidelity serves as a pivotal metric for quantifying the impact of quantum noise on the quality of quantum channels and the reliability of quantum communication. It is defined as the average overlap between the actual final state of a quantum system and the intended target state after undergoing a quantum operation or evolution. The average fidelity is a measure of how well a quantum channel preserves the quantum information and is crucial for assessing the performance of quantum networks. There are existing research works based on fidelity estimation, Ruan [23] proposes a fidelity estimation protocol for entanglement among remote nodes without consideration of quantum measurement errors.

## 2.3 Network Benchmarking

Network benchmarking [8] is an essential process for characterizing quantum networks and ensuring their reliability and performance, which represents an adaptation of the random benchmarking protocol tailored for quantum networks, aligning closely with the characteristics and theoretical underpinnings of such networks [2]. It involves measuring the average fidelity of quantum entanglement links and identifying the optimal operating parameters for a quantum network. Helsen *et al.* [8] proposed a network benchmarking method that is robust to state preparation and measurement errors, allowing for efficient and accurate estimation of link fidelity regardless of how the quantum links are formed. This method is particularly relevant for quantum networks where the fidelity of established entangled links is unknown a priori, and uniform estimation of all links can be costly, especially in networks with numerous links.

Network benchmarking is an essential process to assess the performance and reliability of quantum communication channels. It involves the measurement of key parameters such as the average fidelity of entangled links and the quantum bit error rate (QBER) to characterize the noise resilience of the quantum network.

Let us consider a quantum network composed of $N$ nodes, each capable of generating and sharing entangled pairs of qubits. The network is modeled by a graph $\mathcal{G} = (\mathcal{V}, \mathcal{E})$, where $\mathcal{V}$ represents the set of nodes and $\mathcal{E}$ represents the set of entangled links between nodes. Each link $e_{ij} \in \mathcal{E}$ is associated with a noise channel $\mathcal{N}_{ij}$ that characterizes the quantum noise and decoherence effects along the communication path between nodes $i$ and $j$.

The average fidelity $F_{ij}$ of the entangled link $e_{ij}$ is defined as the average overlap between the intended maximally entangled state $|\Phi^+\rangle$ and the actual final state $\rho_{ij}$ of the qubits after passing through the noise channel $\mathcal{N}_{ij}$. Mathematically, it is given by:

$$F_{ij} = \langle \Phi^+ | \rho_{ij} | \Phi^+ \rangle = \text{Tr}\left( |\Phi^+\rangle \langle \Phi^+ | \rho_{ij} \right), \tag{1}$$

where Tr denotes the trace inner product.

Network benchmarking aims to estimate the average fidelity $F_{ij}$ for each link $e_{ij}$ by performing a series of quantum state tomography measurements. This process requires the transmission of multiple copies of entangled pairs and the subsequent analysis of the measurement outcomes to deduce the properties of the noise channel $\mathcal{N}_{ij}$. In practice, network benchmarking is often limited by the availability of quantum resources and the inherent noise in the system. Therefore, efficient benchmarking protocols are designed to minimize the number of required measurements while still providing accurate estimates of the network's performance.

Helsen *et al.* [8] introduces a resilient network benchmarking technique capable of precisely determining the single quantum link fidelity, which includes two-node and multi-node links. Recent advancements in network benchmarking include Liu *et al.* [13] propose the quantum Border Gateway Protocol (BGP) for entanglement routing across multiple quantum Internet Service Providers (qISPs), integrating network benchmarking with the top-K arm identification problem. Building upon network benchmarking [8], Liu *et al.* [14] develop LINKSELFIE (Link Selection and Fidelity Estimation), an online link selection algorithm integrating concepts from online learning, reducing computation and thus speeding up fidelity estimation by eliminating low-fidelity quantum links.

In conclusion, the field of quantum network architecture and benchmarking is rapidly evolving, with ongoing research focused on developing scalable, efficient, and robust protocols for quantum communication. As quantum networks transition from theoretical constructs to practical implementations, the continued advancement of these foundational concepts will be essential for realizing the full potential of quantum technologies.

## 3 PROBLEM SPACE AND FORMULATION

In this section, we formalize the problem of the quantum network transmission model and noise biases. In practice, two causes of noise changes mainly include spatial and temporal biases.

## 3.1 Quantum Network Transmission Model

We consider a task allocation system denoted as $\mathcal{S} = (\mathcal{D}, \mathcal{L})$, where $\mathcal{D}$ represents the multimedia dataset to be transmitted (which can be decomposed into $m$ quantum states for distribution), and $\mathcal{L}$ represents the available quantum links. Specifically, $\mathcal{D} = \{d_1, d_2, ..., d_m\}$ constitutes a complete multimedia dataset, while $\mathcal{L} = \{l_1, l_2, ..., l_n\}$ denotes the available $n$ quantum links. This study assumes that the multimedia dataset $\mathcal{D}$ is equally divided into $m$ quantum states for distribution, with each quantum state of equal size and consistent transmission time. To focus on multiple quantum states allocation, we do not delve into the effects of switch throughput [7], that is the simultaneous transmission of multiple tasks. Thus, we assume that a quantum link $l_i$ can only transmit one quantum state $d_j$ at a time. This implies that concurrent transmission of two quantum nodes is not considered, aligning with the previous research [8, 14]. At the same time, we consider that the allocated quantum state packets are of the same size.

## 3.2 Constraints and Optimization Goals

Once the task allocation within the quantum network transmission model is determined, the quantum states of the data sub-packets will be transmitted. At this stage, two metrics are used to measure the efficiency of execution: average transmission time and average transmission fidelity. Our goal is to achieve high-fidelity quantum state transmission with as low transmission time as possible.

Achieving high-fidelity transmission while minimizing transmission time involves a trade-off process. To address this trade-off, we formulate a dynamic optimization objective function that balances transmission time and transmission fidelity.

We define two components of the cost model:

**Average Fidelity**:

$$Fidelity = \frac{1}{m} \sum_{i=1}^{m} Fidelity(d_i) \tag{2}$$

Here, $Fidelity(d_i)$ represents the average fidelity of transmitting the $i^{th}$ data sub-packet, computed using the average fidelity formula derived from network benchmarking.

**Cost Time**:

$$Time\_Cost = \frac{1}{m} \sum_{i=1}^{m} (Trans\_Time(d_i) + Estimation\_Time(d_i)) \tag{3}$$

Where $Trans\_Time(d_i)$ denotes the transmission time of the $i^{th}$ data sub-packet, and $Estimation\_Time(d_i)$ denotes the estimation fidelity time of the $i^{th}$ data sub-packet.

**Bounce Number**: it is the process of applying a random Clifford operation on the state from the source node and sending it to the target node, which then performs the same operation and returns it to the source node.

Therefore, the goal of this study is to achieve a balance between transmission fidelity, time overhead, and bounce number based on user-specified trade-off coefficients.

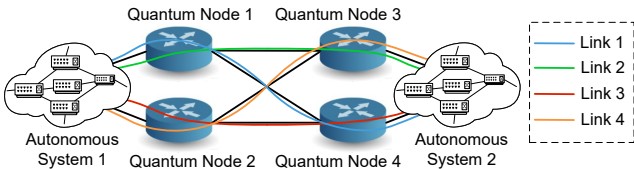

**Figure 2: Quantum network topology and candidate links.**

## 4 DESIGN OF MINERVA

### 4.1 Modeling Quantum Network Links

Consider a network topology consisting of $N$ nodes, where each node represents a critical point within the network, such as a quantum router, a quantum switch, or a quantum repeater. The nodes are interconnected through a set of candidate links that form the communication pathways between the nodes. Let $C$ denote the connection relationship between various nodes, which are potential connections that can be established based on the network's design and requirements. The connectivity between nodes is represented by an adjacency matrix $V$, where $V_{ij}$ is a binary value indicating the presence (denoted as 1) or absence (denoted as 0) of a direct link between node $v_i$ and node $v_j$. The set of all candidate links can be described as $\{c_k\}_{k=1}^{C}$, where each $c_k$ represents a unique link between a pair of nodes.

Given the network topology with $N$ nodes and node connection relationship $C$, we can systematically identify all possible candidate links by performing a unique preorder traversal of the nodes. This method ensures that for each node $v_i$ in the topology, we visit all its predecessors before visiting the node itself, thus generating a sequence of node visits that respects the directed edges without forming any cycles.

The process of generating candidate links can be outlined as follows:

(1) Start at an initial node $v_s$ and mark it as visited.
(2) For the current node $v_i$, iterate through its adjacent nodes $v_j$ in the adjacency matrix $V$.
(3) If $v_j$ has not been visited, perform a preorder traversal on $v_j$ and mark it as visited.
(4) Record the sequence of visited nodes, corresponding to a unique network path.
(5) Repeat the traversal for all unvisited nodes until all nodes have been visited.

By traversing the network in this manner, we can enumerate all unique paths between nodes, which can be considered candidate links. For instance, Figure 2 shows a quantum network topology with 4 nodes between autonomous system (AE) 1 and AE 2. It contains 4 candidate links represented by "blue", "green", "red", and "orange" respectively.

### 4.2 Quantum Link Selection

To select high-fidelity quantum links, a vanilla design refers to performing fidelity measurements on all possible candidate links. However, this is inefficient and unnecessary. Therefore, we identify quantum pathing as a multi-armed bandit problem and leverage the median elimination algorithm [6] to efficiently find high-fidelity

 

links. Specifically, the median elimination algorithm is grounded in the concept of statistical confidence intervals and is tailored to identify an $\varepsilon$-optimal arm with high probability while minimizing the number of trials required. The algorithm operates by iteratively sampling each arm of the bandit and maintaining a running median of the empirical rewards. At each stage, the arm with the lowest empirical reward below the median is eliminated from further consideration. This process continues until an arm that is likely to be $\varepsilon$-optimal is identified, as per the probably approximately correct (PAC) learning framework.

---

**Algorithm 1** Median Elimination Algorithm

---

1: Initialize the set of arms $A$ and calculate the expected rewards $p_i$ for each arm $a_i \in A$.
2: Set the parameters $\varepsilon > 0$ and $\delta > 0$ which determine the desired accuracy and confidence level, respectively.
3: **while** $|A| > 1$ **do**
4:     **for** each arm $a_i \in A$ **do**
5:         Sample arm $a_i$ $n_i$ times to obtain the reward $\hat{p}_i$.
6:     **end for**
7:     Calculate the median empirical reward $\bar{p}$.
8:     **for** each arm $a_i \in A$ **do**
9:         **if** $\hat{p}_i < \bar{p}$ **then**
10:             Eliminate arm $a_i$ from set $A$.
11:         **end if**
12:     **end for**
13: **end while**
14: **return** the remaining arm(s) in set $A$ as the optimal arm(s).

---

The key to the Median Elimination Algorithm's efficiency lies in its ability to reduce the number of necessary samples by eliminating sub-optimal arms early in the process. This not only conserves resources but also accelerates the convergence to an optimal policy. As shown in Algorithm 1, the median elimination algorithm iteratively selects the optimal option from a set of choices by sampling each option, calculating the median reward, and eliminating options with empirical rewards lower than the median. This process continues until only one option remains or until the desired accuracy is achieved.

The algorithm's complexity is analyzed using the PAC model, where the number of samples required is bounded by $O\left(\frac{n}{\varepsilon^2} \log\left(\frac{1}{\delta}\right)\right)$, matching the lower bound derived by the previous work [15]. This bound reflects the trade-off between the number of samples required and the desired level of accuracy and confidence.

The Median Elimination algorithm can be adapted to address the problem of selecting optimal quantum links for transmitting data between quantum nodes. In this context, each quantum link represents an "arm" in the MAB problem, with each link having associated metrics of fidelity and cost time of transmission, analogous to the reward distribution of a bandit's arm. Model the quantum network and initialize the exploration state table of the quantum network. Just like in the MAB problem, each quantum link (QL) is "pulled" a certain number of times to collect data on its performance (*i.e.,* fidelity) and recorded. After each round of sampling, the median empirical fidelity and transmission cost time are calculated. QLs whose empirical fidelity was below the median and therefore

considered suboptimal were eliminated. Specifically, we aim to find the median fidelity $\mu$ of the link set $\mathcal{L}$. Let $l_{\text{fid}}$ be the fidelity of all links in $\mathcal{L}$. To calculate the median fidelity $\mu$, we first sort the probabilities $l_{\text{fid}}$ in ascending order:

$$l_{\text{fid}}^{(1)} \leq l_{\text{fid}}^{(2)} \leq \ldots \leq l_{\text{fid}}^{(N_l)} \tag{4}$$

where $N_l$ is the total number of links in $\mathcal{L}$. If $N$ is odd, the median $\mu$ is the value of the middle element:

$$\mu = l_{\text{fid}}^{\left(\frac{N_l+1}{2}\right)} \tag{5}$$

If $N$ is even, the median $\mu$ is the average of the two middle elements:

$$\mu = \frac{1}{2}\left(l_{\text{fid}}^{\left(\frac{N_l}{2}\right)} + l_{\text{fid}}^{\left(\frac{N_l}{2}+1\right)}\right) \tag{6}$$

The sampling and elimination process is repeated until a single QL is retained or until the remaining QL is sufficiently optimal (within $\epsilon$ of the best performance). The remaining QL can then be selected as the optimal transmission path.

## 4.3 Multimedia Chunk Profit Modeling

In the context of quantum networks, the task allocation involves distributing $m$ multimedia chunks between two quantum nodes using $n$ available quantum links (QLs). Each QL is characterized by a fidelity $F_i$ and a cost time $C_i$, which represent the reliability and the time required to transmit a multimedia chunk, respectively. The objective is to allocate the multimedia chunks across the QLs to maximize the overall performance, defined by the profit function:

$$\text{Profit} = \alpha \times \text{Fidelity} - \beta \times \log(\text{Cost Time}) - \gamma \times \log(\text{Bounce Num}) \tag{7}$$

where $\alpha$, $\beta$, and $\gamma$ are weights that reflect the relative importance of fidelity, cost time, and bounce num in the specific quantum network application, respectively.

To mathematically define the problem, let $\mathcal{M} = \{m_1, m_2, \ldots, m_k\}$ be the set of multimedia chunks and $\mathcal{L} = \{l_1, l_2, \ldots, l_n\}$ be the set of quantum links. The transmission policy $\mathcal{P}$ is a mapping from $\mathcal{M}$ to $\mathcal{L}$, *i.e.,* $\mathcal{P} : \mathcal{M} \to \mathcal{L}$. The fidelity $F_i$, cost time $C_i$, and bounce number $B_i$ for each QL $l_i$ are considered as constraints in the allocation problem.

The optimization problem can be formulated as:

$$\text{Maximize: } \sum_{m \in \mathcal{M}} \left(\alpha \times F_{\mathcal{P}(m)} - \beta \times \log(C_{\mathcal{P}(m)}) - \gamma \times \log(B_{\mathcal{P}(m)})\right)$$

$$\text{Subject to: } \mathcal{P}(m) \in L \text{ for all } m \in \mathcal{M}$$
$$A(m_1) \neq \mathcal{P}(m_2) \text{ for any distinct } m_1, m_2 \in \mathcal{M}$$

## 4.4 Multimedia Chunk Transmission Strategy

**Initialization.** The algorithm commences with the initialization of the multimedia chunk set, denoted as $\mathcal{M}$. A set of parameters, $\alpha$, $\beta$, $\gamma$, $\mathcal{L}$, and $\mathcal{M}$, is defined to govern the allocation process. The quantum links, inclusive of multiple repeaters, are established to facilitate the transmission of data. (i) The corpus of multimedia chunks destined for allocation is initialized, and represented as $\mathcal{M}$. (ii) The algorithmic parameters, namely $\alpha$, $\beta$, $\gamma$, $\mathcal{L}$, and $\mathcal{M}$, are delineated and assigned. (iii) Quantum links, integrated with a network of repeaters, are configured to enable efficient data transmission.

**Iterative Allocation.** The allocation procedure is executed iteratively for each multimedia chunk $m_i$ within the set $\mathcal{M}$.

(1) An empty set of links is initialized for the allocation of multimedia chunk $m_i$.
(2) The initial profit for each viable link is computed, taking into account the potential allocation of $m_i$.
(3) The link with the maximal initial profit is identified and selected.
(4) Data packet $m_i$ is allocated to the chosen link.
(5) The fidelity, cost time, and bounce number parameters for the allocated link are updated.
(6) The profit for all links is recalculated in light of the updated parameters.
(7) This iterative process is reiterated until all multimedia chunks are effectively allocated.

**Profit Calculation.** The computation of fidelity is performed incrementally and in real-time during the link selection phase to mitigate preliminary computational expenditures. Concurrently, the cost time and bounce number are incremented as packets are transmitted across the allocated links. Fidelity is dynamically assessed during the link selection process to minimize preliminary computational overhead. The cost time and bounce number are incremented as packets traverse through the allocated links.

**Allocation Termination.** The allocation process is terminated upon the successful allocation of all multimedia chunks to quantum links. Furthermore, the allocation strategy is outputted, detailing the assigned links and their corresponding multimedia chunks.

## 5 EVALUATION

### 5.1 Experiment Setup

**Testbed.** We conduct simulations of quantum networks using the Netsquid library with the network benchmarking [8]. Each bounce of every link is simulated 20 times to ensure statistical reliability. The simulations are performed on a machine equipped with an Intel(R) Core(TM) i7-9700 CPU @ 3.00GHz and 32GB RAM.

**Multimedia Chunk Sizes.** As for the multimedia chunk size, we adopt the representative network traces [22] involving 460 traces from Norway's 3G HSDPA, which is also used in the evaluation of previous work about streaming media transmission [1, 16, 26]. We randomly sample from those traces as the size of each multimedia chunk.

**Quantum State Groups.** Due to limitations imposed by current NISQ-era quantum technologies, we utilize simulated quantum states to represent our multimedia chunk data. Each chunk of data is of equal size. We simulate three quantum link scales, $\mathcal{L} = \{5, 10, 20\}$, and the number of multimedia chunks in each group, refers to $\mathcal{M} = \{20, 50, 80, 100\}$. The topology of the quantum network is not considered in this paper, and transmission is assumed to occur between multiple paths within two quantum nodes.

**Transmission Algorithm Baselines.** Three methods are employed for comparison:

- **Random:** Transmission allocation is randomly distributed across quantum links, without consideration for fidelity or cost time.

- **Uniform:** Allocation is conducted in a uniform manner, ensuring an equal number of transmissions across all quantum links, independent of their individual characteristics.
- **Greedy:** Utilizes a greedy algorithm that prioritizes transmission through quantum links with the highest fidelity, aiming to maximize the overall system performance based on the individual link quality.

### 5.2 Transmission Performance Evaluation

In this section, we conduct an evaluation of Minerva's transmission performance compared to three baseline strategies, employing a fixed data chunk size denoted by $\mathcal{M} = 100$. The experimental result is shown in Table 1, which presents the average results across 100 chunks of multimedia quantum states. Metrics such as "Bounces," "Cost time," "Fidelity," and "Profit" denote the average bounce number of quantum links, the average transmission time, the average fidelity of quantum links, and the overall profit, respectively. We assess these metrics across three distinct strategies: *Random*, *Uniform*, *Greedy*, and our proposed method (*Minerva*), tested over quantum link configurations $\mathcal{L} = \{5, 10, 20\}$.

Overall, the Random strategy exhibits the poorest profit performance due to its disregard for the intrinsic characteristics of quantum links. It consistently yields the highest bounce numbers but correspondingly the lowest profit, especially evident with increased quantum link counts. This indicates the inefficiency of random allocation, failing to capitalize on higher fidelity links. Conversely, the Uniform strategy, distributing transmissions evenly across all links, demonstrates more consistent performance across different link counts. While uniformly distributed methods generally incur lower execution times, they sacrifice fidelity for speed, resulting in comparable profit levels to random methods, albeit with higher bounce numbers. The Greedy strategy, prioritizing links based on individual fidelity, shows notable improvement over previous approaches, yielding the highest fidelity and substantial profit across all link counts except for the 5-link configuration where it exhibits slightly lower fidelity but maintains high profit. However, the inherent design of the Greedy algorithm, focusing solely on maximizing fidelity without considering bounce numbers or execution time, makes it challenging to achieve optimal overall performance, leading to performance levels comparable to Random and Uniform methods, yet still significantly trailing our proposed approach. Our method, employing a fidelity-aware allocation strategy, effectively balances fidelity, cost-time, and overall profit considerations. It consistently outperforms baseline methods across all evaluation metrics, achieving low computation times and bounce numbers while maximizing fidelity and profit. This underscores its ability to effectively navigate the trade-offs among these influencing factors.

In summary, our results highlight the efficacy of our *Minerva* method in optimizing quantum data chunk transmission. Strategic selection of quantum links based on fidelity and cost-time characteristics significantly outperforms traditional random and uniform distribution strategies, as well as the Greedy approach, particularly in scenarios with higher quantum link counts.

 

**Table 1: Performance evaluation of multimedia quantum data chunks ($\mathcal{M} = 100$) across various numbers of quantum links. Among them, the unit of "Bounces" refers to $1e^4$.**

| Links | Link scales $\mathcal{L}$ = 5 | | | | Link scales $\mathcal{L}$ = 10 | | | | Link scales $\mathcal{L}$ = 20 | | | |
|---|---|---|---|---|---|---|---|---|---|---|---|---|
| Method | Bounces | Cost time | Fidelity | Profit | Bounces | Cost time | Fidelity | Profit | Bounces | Cost time | Fidelity | Profit |
| Random | 75.23 | 35.33 | 0.82 | 38.86 | 84.53 | 16.57 | 0.85 | 48.82 | 83.21 | 8.67 | 0.83 | 54.02 |
| Uniform | 88.54 | 20.75 | 0.85 | 45.87 | 85.97 | 10.89 | 0.80 | 48.19 | 86.19 | 5.44 | 0.81 | 56.70 |
| Greedy | 73.16 | 40.68 | 0.90 | 45.63 | 71.74 | 18.62 | 0.92 | 56.33 | 72.89 | 11.98 | 0.93 | 61.95 |
| **Ours** | **65.61** | 38.36 | **0.95** | **52.45** | **64.68** | 21.91 | **0.97** | **60.69** | **59.86** | 9.65 | **0.96** | **69.43** |

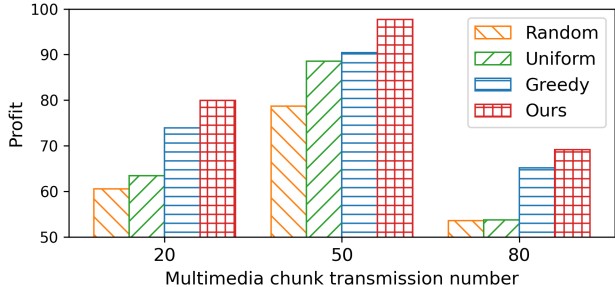

**Figure 3: Profit calculation for different multimedia chunk sizes with quantum links $\mathcal{L}$ = 10.**

## 5.3 Comparison of Transmission Profit under Different Multimedia Chunk Groups

We investigate the profit performance of various multimedia chunk transmission sizes under a fixed number of quantum links $\mathcal{L}$ = 10, as depicted in Figure 3. The profit metric is plotted against the number of transmitted multimedia chunks for four distinct algorithms: Random, Uniform, Greedy, and our proposed method (referred to as "Ours"). The overall profit exhibits an initial increase followed by a decline as the number of transmitted packets varies from 20 to 50 and then to 80. This trend suggests that with a fixed number of quantum links ($\mathcal{L}$ = 10), the system gradually approaches the upper limit of its data transmission capacity as the packet volume increases, without considering the parallel transmission of multiple packets.

Specifically, the Random and Uniform algorithms demonstrate relatively lower profit performance across varying transmission volumes. Their profit increment is modest with increasing transmission numbers, reflecting the stochastic nature of these approaches. In contrast, the Greedy algorithm showcases significant profit enhancement, particularly at lower transmission volumes, leveraging the highest fidelity links for rapid profit escalation. However, as the transmission volume increases, the rate of profit growth appears to diminish, indicating a potential saturation point where further exploitation of high-fidelity links yields diminishing returns. Our proposed algorithm consistently outperforms the others, achieving the highest profit across all transmission volumes. Its optimized selection and allocation of quantum links afford a substantial profit advantage over alternative strategies. The steep ascent of the profit trend line for the Ours algorithm underscores its efficiency and effectiveness in managing multimedia chunk transmissions. In summary, the experimental results presented in Figure 3 emphasize the superior profit-maximizing capability of our proposed algorithm

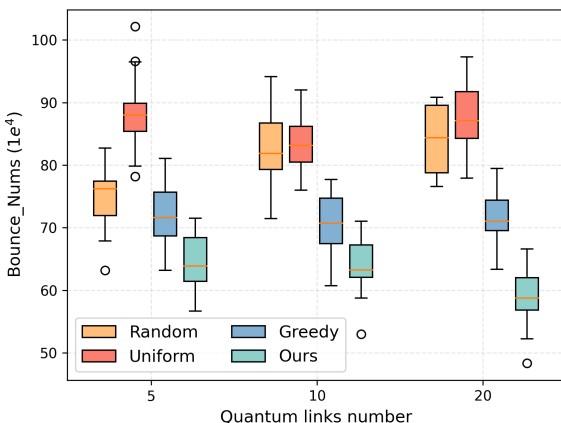

**Figure 4: Bounces number of different quantum links with multimedia chunk size $\mathcal{M}$ = 50.**

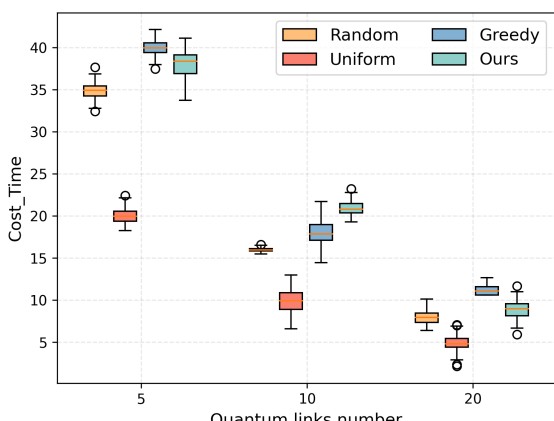

**Figure 5: Transmission cost time of different quantum links with multimedia chunk size $\mathcal{M}$ = 50.**

for multimedia chunk transmissions across varying quantum link counts.

## 5.4 Detailed Profit Analysis

To delve into the variation of profit's constituent elements across different quantum links, namely *bounce number*, *cost time*, and *fidelity*, we conduct tests on multimedia chunk size ($\mathcal{M}$ = 50) packets under three scenarios of quantum link counts $\mathcal{L}$ = {5, 10, 20}. The experimental outcomes are depicted in Figures 4, 5, and 6.

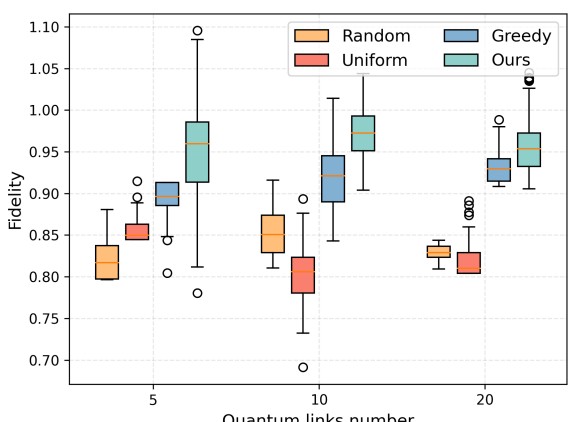

**Figure 6: Average fidelity of different quantum links with multimedia chunk size $M = 50$.**

**Bounce Numbers for Quantum Links** Figure 4 illustrates the bounce number, measured in units of $10^4$. The observed trend is generally as follows: *uniform > random > greedy > ours*. This ordering becomes more pronounced with increasing quantum links, as our algorithm exploits larger optimizable spaces compared to others.

**Cost Time for Quantum Links** From Figure 5, it's evident that overall data transmission time decreases notably with increasing quantum links. Regarding the change in transmission time for each set of quantum links, the ordering typically follows: *uniform < ours < greedy < random*. The uniform distribution, prioritizing speed over transmission quality, achieves the shortest execution time, with ours slightly behind but ensuring transmission quality.

**Fidelity for Quantum Links** Figure 6 demonstrates that although Uniform evenly distributes tasks, it overlooks transmission quality, resulting in the lowest average fidelity. As quantum link counts increase, fidelity generally trends upwards, with our method consistently outperforming other baseline methods.

In summary, the experimental results across all three figures consistently highlight the efficacy of our strategy in optimizing hop count and fidelity while minimizing cost and time. Our approach offers a robust and efficient algorithm for quantum link selection and task assignment. Conversely, the Random strategy performs poorly across all metrics, while Uniform achieves the shortest transmission time but sacrifices transmission quality. Though the Greedy algorithm shows some capacity improvement, a general greedy approach struggles to balance the three metrics optimally. These findings underscore the critical role of strategy selection and allocation in quantum network optimization.

## 6 DISCUSSION AND FUTURE WORK

**Advantages of Quantum Networks.** Quantum networks offer several advantages over traditional networks when it comes to multimedia transmission, stemming from the unique properties of quantum mechanics. For example, (i) enhanced security: quantum networks leverage quantum encryption methods, such as quantum key distribution (QKD), which provide a level of security that is fundamentally impossible to achieve with classical networks. (ii) Immunity to eavesdropping: due to the no-cloning theorem in

quantum mechanics, it is impossible to create an exact copy of an unknown quantum state without disturbing the original state. This means that an eavesdropper cannot intercept and copy the quantum information without being noticed, providing inherent protection for the transmitted multimedia content. (iii) Potential for quantum data compression: quantum networks could, in theory, employ quantum data compression techniques that can compress and decompress information more efficiently than classical methods. This could lead to more efficient transmission of multimedia data, requiring less bandwidth and reducing transmission times.

**Practicality and Extensibility.** In our design framework, the fidelity term in the profit function reflects the quality of service (QoS) in data transmission. The proposed algorithms in Minerva can efficiently optimize fidelity by dynamically adjusting data routing and resource allocation, ensuring superior user experience in multimedia applications. Meanwhile, the algorithm prioritizes minimizing time and bounce, enhancing network efficiency, and reducing latency. Moreover, the scalability of the proposed algorithm is promising for accommodating the evolving demands of multimedia networks. Quantum-inspired heuristics and hybrid classical-quantum approaches may offer viable solutions to address scalability concerns while ensuring adaptability to dynamic network environments.

**Limitations and Future Works.** Our work has some limitations. Firstly, the fidelity of quantum devices arises from diverse factors, such as hardware manipulations and environmental conditions, and their interplay is intricate. Tailor-made designs may be a promising direction for different quantum chips. In the future, we could also consider optimizing the topology of quantum networks to achieve high-fidelity multimedia transmission.

## 7 CONCLUSION

In this paper, we study the multimedia data resource allocation problem in quantum networks. To solve this problem, we formalize quantum fidelity estimation and link selection as an optimal arm identification problem and design an efficient algorithm Minerva, a new framework for improving quantum network performance through link selection and transmission strategies, using median elimination to estimate the fidelity and to select quantum links for each multimedia block transmission. To optimize the transmission of multimedia blocks in a quantum network, we can employ a scheduling strategy to maximize the cumulative benefits of block transmission while considering link fidelity and overall network utilization. Simulation results show that Minerva can simultaneously guarantee a smaller bounce number and time overhead in different scenarios, while also achieving the highest average fidelity transmission and guaranteeing the quality of data transmission.

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
