# OpenReview forum: "Minerva: Enhancing Quantum Network Performance for High-Fidelity Multimedia Transmission"
_acmmm.org/ACMMM/2024/Conference — MM2024 Poster_

### Official Review · Reviewer_zbFg · 2024-05-06

**Rating:** 1
**Confidence:** 1

**Summary:**

The paper is on so called quantum networks, a vision to connect quantum, computers by networks able to transmit qbits.
To be relevant for MMSys the authors chose chunked video as a theme example and defined a profit that is related to the transmission of such chunks and that the algorithm tries to maximize.

**Strengths:**

I'm not at all an expert on quantum networks.
I cannot extract any comprehensible output and insight from the paper, so I cannot state any real strength.

**Limitations:**

For me, the underlying assumptions and math to judge the network quality are relatively trivial (Eq. (1&ndash;3) for the fidality and Eq.(7) for the overall profit) and can be applied completely outside the context of quantum networks and chunked video transmission. The paper boils down to a (relatively simple) maximization problem.
The comparison is not done with any other proposal but just with *random* and *greedy* usage of the available set of links, which again just argues within the maximization problem itself and shows that the proposed link selection scheme outperforms those very rude schemes (planning is better than randomness...).
For me not surprising.

**Suitability:**

1

---

### Official Review · Reviewer_35w3 · 2024-05-23

**Rating:** 5
**Confidence:** 3

**Summary:**

This paper is an innovative work and it is well written. It presents Minerva, a framework to enhance quantum network performance for high-fidelity multimedia transmission. Minerva makes link selection and fidelity evaluation strategies for each multimedia chunk transmission using a best-arm identification algorithm with median elimination. It aims to maximize the cumulative benefit of transmissions while ensuring the link fidelity and overall network utilization.

**Strengths:**

1.	This is a compelling work that creatively applies quantum network technology to traditional multi-media video transmission, enhancing the transmission performance transmission and breaking through the original bottleneck. This provides a fresh approach of thinking about the field of quantum networks and greatly facilitates their applicability.
2.	This paper is well written and it is excellent in terms of organization and language, some minor issues will be listed later.
3.	The authors conducted a number of experiments to verify the effectiveness of Minerva, and in particular, they describe the details of their experiments well.

**Limitations:**

1.	Minerva shows good scalability and practicality through block-level transmission. However, the transmission capacity of quantum networks is limited, and further consideration of more fine-grained transmission in the future is necessary to further exploit the potential of quantum video transmission.
2.	The representation of acronyms should be uniform, e.g., initial letters in all upper or all lower case, which is currently mixed.
3.	The authors should have emphasised "Minerva" in the abstract, otherwise it would have been confusing to read later.
4.	Although the authors provide detailed background and related work about their study, I would recommend separating the two to provide a better understanding.
5.	The description of the metrics in Section 3.2 should be clearer, for example, the authors present two cost components, but three are given below.

**Suitability:**

3

---

### Official Review · Reviewer_m75N · 2024-05-29

**Rating:** 4
**Confidence:** 2

**Summary:**

The paper introduces a novel framework aimed at improving the performance of quantum networks for multimedia data transmission with a focus on high fidelity. The authors propose a method to address the challenge of ensuring high-fidelity transmission over quantum links by formalizing the quantum fidelity estimation and link selection as an optimal arm identification problem. They employ a median elimination strategy to estimate fidelity and select the best quantum link for multimedia chunk transmission. The framework also includes a scheduling strategy to optimize the transmission of multimedia chunks, taking into account link fidelity and overall network utilization. The paper presents extensive experimental results that demonstrate the advantages of the proposed Minerva framework over randomized and uniformly distributed methods in terms of reduced bounce number, execution time, and improved average fidelity.

**Strengths:**

Innovative Approach: The paper presents a unique approach to quantum network optimization by leveraging the multi-armed bandit problem and median elimination algorithm for link selection, which is a novel contribution to the field.

Problem Formulation: The authors have clearly defined the problem space and formulated the quantum network transmission model and noise biases effectively.

Algorithm Design: The design of the Minerva algorithm is comprehensive, with a clear explanation of the modeling of quantum network links, quantum link selection, multimedia chunk profit modeling, and transmission strategy.

Performance Evaluation: The comparison of the proposed method with existing strategies (Random, Uniform, Greedy) is thorough and demonstrates the superiority of the Minerva framework in various metrics.

Potential Impact: The work has the potential to significantly impact the field of quantum networking and multimedia transmission by providing a more efficient and reliable method for data transfer.

**Limitations:**

Practical Implementation: While the paper presents a strong theoretical framework, there is a lack of discussion on the practical implementation challenges and the scalability of the proposed system to real-world quantum networks.

Complexity Analysis: The paper could benefit from a deeper analysis of the computational complexity of the proposed algorithms, especially in terms of scalability and resource requirements.

Diversity of Quantum Networks: The evaluation does not consider a diverse range of quantum network topologies, which could impact the generalizability of the results.

Security Considerations: Although the paper touches on the security advantages of quantum networks, a detailed analysis of potential security vulnerabilities and countermeasures within the proposed framework is missing.

Hardware Dependency: The fidelity and performance of quantum devices are heavily dependent on hardware specifics, which are not extensively discussed. The paper could benefit from considering the impact of different quantum hardware on the proposed system.

Generalization to Other Domains: The paper primarily focuses on multimedia transmission. It would be beneficial to see if the proposed methods can be generalized to other types of data or applications.

**Suitability:**

3

---

### Official Review · Reviewer_PKG8 · 2024-05-31

**Rating:** 2
**Confidence:** 2

**Summary:**

The proposed framework, Minerva, focuses on enhancing the fidelity and efficiency of quantum links used for transmitting multimedia chunks. It achieves this by formalizing the problem of quantum fidelity estimation and link selection as a best-arm identification problem, employing a median elimination strategy. The approach also includes a dynamic scheduling strategy to maximize the cumulative benefit of chunk transmissions while maintaining high fidelity and efficient network utilization.

**Strengths:**

1.	The paper introduces an innovative framework that leverages a median elimination strategy for link selection, which is a novel application in the context of quantum networks.
2.	The extensive experiments and comparative analysis against randomized and uniformly distributed methods provide strong empirical evidence of the framework's effectiveness, demonstrating clear improvements in performance metrics such as bounce number, execution time, and fidelity.

**Limitations:**

1.	The scope of this paper is weakly related to multimedia. Although the context of this paper is video streaming, the major design focuses heavily on networking instead of the media itself.
2.	While the proposed method shows significant improvements, the complexity of implementing such a framework in real-world quantum networks might pose challenges. The scalability of the approach, especially in larger and more complex quantum networks, is not thoroughly addressed.
3.	There is a lack of detailed discussion on the practical aspects of deploying Minerva in existing quantum networks. Considerations such as hardware requirements, integration with current technologies, and potential barriers to implementation are not sufficiently covered.

**Suitability:**

1

---

### Official Review · Reviewer_GK7W · 2024-06-07

**Rating:** 4
**Confidence:** 4

**Summary:**

This paper proposes a new link selection algorithm for quantitative networks. This algorithm considers link fidelity, transmission time, and bounce num comprehensively to achieve the maximum benefit of link decisions. The paper conducts simulation experiments to verify the effectiveness of the algorithm. In conclusion, the definition and description of the problem in the paper are clear, and the solution is reasonable, but the research on related work and the comparative experiment section are slightly insufficient.  It is suggested that this paper is borderline accept.

**Strengths:**

- The writing of the paper is good, the sentences are smooth, and the definition and description of the problem are clear and easy to understand.
- The design is reasonable. The work proposes a link selection algorithm that integrates multiple optimization objectives, which has a better effect than algorithms that only consider a single dimension.

**Limitations:**

- The description of the related work section is relatively weak, lacking an analysis of the problems and limitations of the existing work, such as what problems exist in [14], which is also a link selection algorithm. It is recommended to supplement this part of the analysis in Chapter 2.
- The experimental evaluation section is relatively weak. The authors choose some heuristic algorithms as the baseline in the selected experiments, which is beneficial to compare the benefits of the paper's work, but it is not enough to show the advancement of the work compared to the state-of-the-art work. Comparative experiments with related work should be added.
- In the profit function which is used to evaluate the algorithms (4.3), the weight parameters also have a significant impact on the final benefit. The author did not explain how the parameters alpha, beta, and gama are set and whether the setting is reasonable. At the same time, the experimental section should also compare the experimental results under different weight configurations.

**Suitability:**

3

---

### Meta-Review · Area_Chair_hRwg · 2024-07-02

**Recommendation:** Accept (Poster)
**Confidence:** 2

**Metareview:**

This paper studies the multimedia data resource allocation problem in quantum networks. The authors formalize quantum fidelity estimation and link selection as an optimal arm identification problem and design an efficient algorithm, using median elimination to estimate the fidelity and to select quantum links for each multimedia block transmission.
We have collected comments from the reviewers and we hope these comments are useful to improve the paper.